# Structural and dynamic mechanisms of CBF3-guided centromeric nucleosome formation

Ruifang Guan[1,6], Tengfei Lian[2,6], Bing-Rui Zhou [1], Emily He[3], Carl Wu [4], Martin Singleton [5] & Yawen Bai [1✉]

Accurate chromosome segregation relies on the specific centromeric nucleosome–kinetochore interface. In budding yeast, the centromere CBF3 complex guides the deposition of CENP-A, an H3 variant, to form the centromeric nucleosome in a DNA sequence-dependent manner. Here, we determine the structures of the centromeric nucleosome containing the native CEN3 DNA and the CBF3core bound to the canonical nucleosome containing an engineered CEN3 DNA. The centromeric nucleosome core structure contains 115 base pair DNA including a CCG motif. The CBF3core specifically recognizes the nucleosomal CCG motif through the Gal4 domain while allosterically altering the DNA conformation. Cryo-EM, modeling, and mutational studies reveal that the CBF3core forms dynamic interactions with core histones H2B and CENP-A in the CEN3 nucleosome. Our results provide insights into the structure of the budding yeast centromeric nucleosome and the mechanism of its assembly, which have implications for analogous processes of human centromeric nucleosome formation.

[1] Laboratory of Biochemistry and Molecular Biology, National Cancer Institute, National Institutes of Health, Bethesda, MD, USA. [2] Laboratory of Membrane Proteins and Structural Biology, Biochemistry and Biophysics Center, National Heart, Lung, and Blood Institute, National Institutes of Health, Bethesda, MD, USA. [3] John A. Paulson School of Engineering and Applied Sciences, Harvard University, Cambridge, MA, USA. [4] Department of Biology, Johns Hopkins University, Baltimore, MD, USA. [5] Structural Biology of Chromosome Segregation Laboratory, The Francis Crick Institute, London, UK. [6] These authors contributed equally: Ruifang Guan, Tengfei Lian. ✉email: baiyaw@mail.nih.gov

Centromeres mediate the attachment of chromosomes to the mitotic spindle by the kinetochore complex that binds microtubules, which is responsible for accurate chromosome segregation during mitosis[1,2]. Mis-segregation of chromosomes can lead to aneuploidy, a hallmark of cancer[3]. Centromeres are marked by specific nucleosomes in which the canonical histone H3 is replaced by the CENP-A variant[4,5]. Human centromeres are regional, including megabase DNA with repeats of two alternating ~171 base pairs (bp) α-satellite DNA sequences[6–8]; one of them consists of the 17 bp CENP-B box DNA motif that is specifically recognized by the centromere CENP-B protein[9,10]. Studies using native chromatin immunoprecipitation followed by sequencing of CENP-A-containing particles reveal that the octamer is the major form of CENP-A nucleosomes in normal centromeres and naturally occurring neocentromeres[11]. Structures of human CENP-A nucleosome core particle containing one half human palindromic α-satellite DNA, Widom "601" DNA, or a native α-satellite DNA have been solved at near-atomic/atomic resolution using X-ray crystallography and cryo-EM[12–17], respectively. Moreover, the structure of the human CENP-A nucleosome containing the native α-satellite DNA is determined at 2.6 Å resolution using a single-chain antibody fragment (scFv)-assisted cryo-EM method in which each nucleotide can be resolved[18].

In contrast, budding yeast chromosomes have point centromeres with single nucleosomes that are defined by conserved ~125 nucleotide segments, including three centromere determining elements (CDEs): CDEI (8 bp), CDEII (~80–90 bp, and AT-rich), and CDEIII (~25 bp)[19–22]. The cryo-EM structure of budding yeast CENP-A (Cse4 or CENP-A$^{Cse4}$) nucleosome containing the Widom 601 nucleosome positioning DNA was determined at 2.7 Å resolution[23]. However, the structure of the nucleosome containing a native centromeric DNA sequence remains unavailable. Instead, two structural models have been proposed for the nucleosome containing CEN3 DNA, which have different dyad positions[24,25]. Also, some non-octameric forms of centromeric nucleosomes have been proposed to exist in vivo[26–29].

The S. cerevisiae centromere binding factor 1 and 3 (CBF1 and CBF3) bind to CDEI and CDEIII, respectively, in a specific sequence-dependent manner. CBF1 with a helix-loop-helix structure is required for chromosome stability[30,31]. CBF3 is a four-protein complex consisting of subunits Ndc10, Skp1, Ctf3, and two copies of Cep3[25,32–34]. CBF3 assembles on CEN loci by engaging its Gal4 domain in one of the two Cep3 subunits with the CCG motif in CDEIII[25,34,35]. The Ndc10 subunit binds to DNA in a sequence-independent manner and associates with CBF1 and Scm3[36], a specific chaperone for Cse4[5,37,38]. Both CBF3 and the CCG motif play essential roles in chromosome segregation[39] and the assemble and function of CBF3 are highly regulated[40,41].

Two structural mechanisms have been proposed for how CBF3 may engage with the CEN DNA and guide the formation of the centromeric nucleosome. The initial model suggests that the Ndc10 dimer binds to CEN DNA in a defined register through its interactions with CBF1 and CBF3, which bring CDEI and CDEIII together to form a loop. Subsequently, the Scm3–Cse4–H4 heterotrimeric complex is recruited through Scm3–Ndc10 interaction[36]. It is shown recently that Ndc10 only weakly associates with the CBF3core (or CBF3core, consisting of Skp1, Ctf3, and 2xCep3) and the CBF3core plays a role in DNA bending[32,33]. In contrast, based on the cryo-EM structure of the CBF3 bound to CDEIII DNA in which the Ndc10 dimer binds to the same DNA fragment[25], it is proposed that the proximity of Ndc10 to the CENP-A$^{Cse4}$ protein in the CEN nucleosome could provide the mechanism for how CBF3 would recruit CENP-A$^{Cse4}$

nucleosome to CEN loci. In this proximity model, CDEI, CDEII, and more than 20 bp DNA before CDEI interact with core histones to form the CEN3 CENP-A$^{Cse4}$ nucleosome core while CDEIII serves as linker DNA bound to the CBF3 dimer. However, earlier in vivo nucleosome mapping studies by MNase digestion of the CEN centromeres have shown that no significant additional regions beyond the CDEs are protected[20,42,43].

## Results

To understand the mechanism of the centromeric nucleosome formation guided by CBF3, we first tried to use the single-particle cryo-EM method to determine the structure of the CEN3 CENP-A$^{Cse4}$ nucleosome. We reconstituted the CENP-A$^{Cse4}$ nucleosome using a 136 bp DNA that includes the native CEN3 sequence (Fig. 1a)[44]. However, the reconstituted nucleosome dissociated on the cryo-EM grid. Attempt to use chemical cross-linking to fix the nucleosome was unsuccessful. The dissociation is likely caused by the surface tension at the water–air interface[45], and the weaker binding of the CEN3 DNA to the core histone as its AT-rich DNA is less bendable and unfavorable for wrapping the core histones[46]. To overcome this problem, we used a single-chain antibody fragment (scFv) that was shown previously to bind to the core histone H2A–H2B in the nucleosome and prevented the nucleosome from dissociation during the blotting–freezing process[18]. In the presence of the antibody, we were able to observe intact particles of the CEN3 CENP-A$^{Cse4}$ nucleosome bound to scFv and obtained a density map of the nucleosome–scFv$_2$ complex at 3.1 Å resolution (Fig. 1b, Supplementary Fig. 1a–f, and Table 1). The high quality of the density map allowed us to build a structural model of the nucleosome with a uniquely positioned DNA (Fig. 1c, d and Supplementary Fig. 1g–j).

In the structure, we found that 115 bp DNA, including CDEII (83 bp), CDEIII (26 bp), and 6 bp of CDEIII$^R$ (the region on the right side of CDEIII) (Fig. 1a), interacted with the core histones and formed a well-defined left-handed ~1.3-turn super-helical structure. The end regions of the CEN3 DNA (2–14 and 131–137) showed much weaker density, indicating that they have flexible conformations. In comparison, the recently reported structure of the CENP-A$^{Cse4}$ nucleosome core containing the non-native Widom 601 DNA includes 119 bp structured DNA, even though the core histones in both of the CEN3 and 601 CENP-A$^{Cse4}$ nucleosomes show similar structures[23] with a root mean square deviation (RMSD) of 0.9 Å (Supplementary Fig. 1k). In contrast, the human CENP-A nucleosome core particle containing the native α-satellite DNA includes 145 bp[18]. Amino acid sequence alignment of the human CENP-A and budding yeast CENP-A$^{Cse4}$ shows that the lack of several positively charged residues in the αN helix of CENP-A$^{Cse4}$ is the likely cause for the flexible DNA ends in the CEN3 CENP-A$^{Cse4}$ nucleosome (Fig. 1e). Also, the DNA position in our cryo-EM structure of the CEN3 nucleosome is different from those in the earlier two structural models[24,25].

In the structure of the CEN3 CENP-A$^{Cse4}$ nucleosome, the CCG motif, located between the super-helical locations 3 and 4 of the DNA, is accessible for binding by the Gal4 domain (Fig. 1c), suggesting that the CBF3core could bind to the CEN3 nucleosome. To test it, we conducted the nucleosome binding study using the CBF3core purified from budding yeast cells after treating it with the phosphatase (bacteria lambda protein). Previous studies have shown that the purified CBF3core are phosphorylated, and dephosphorylation is required for CBF3core binding to CEN3 DNA[32]. Indeed, we found that the CBF3core but not the purified phosphorylated form could bind to the CEN3 CENP-A$^{Cse4}$ nucleosome in an electrophoretic mobility shift

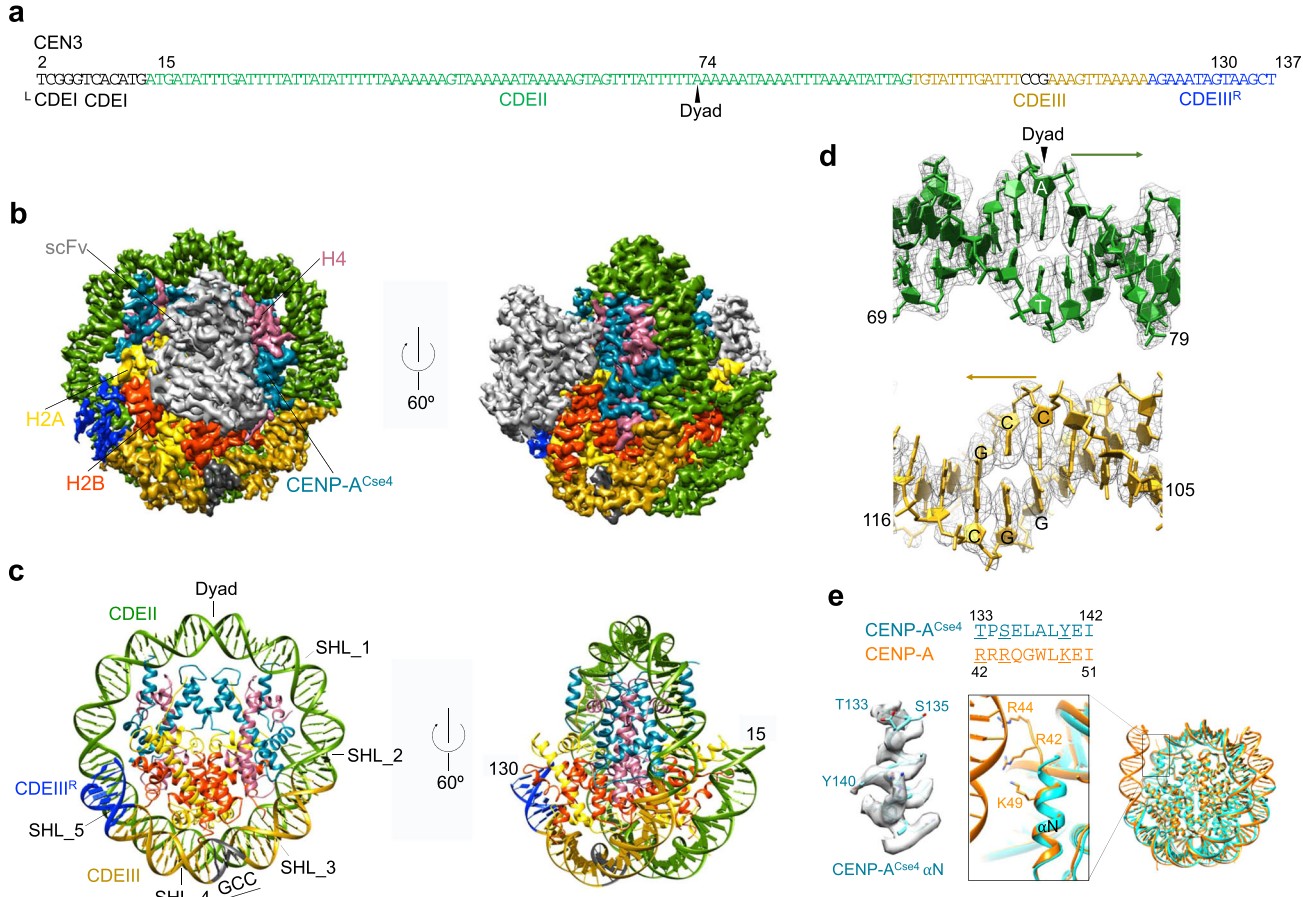

**Fig. 1 Cryo-EM Structure of the CEN3 CENP-A$^{Cse4}$ Nucleosome. a** The CEN3 DNA sequence with highlighted specified regions. **b** Density maps of the CEN3 CENP-A$^{Cse4}$ nucleosome bound to scFv in top (left) and 60° rotational (right) views. **c** The structures of the CEN3 CENP-A$^{Cse4}$ nucleosome with the views as in **b**. **d** Density maps and structures for the dyad and CCG regions. Arrows indicate the direction of the DNA sequence as in **a**. **e** Density map of the αN helix of CENP-A$^{Case4}$ in the CEN3 nucleosome (cyan) and structural comparison with the corresponding region in human CENP-A nucleosome (orange, PDB: 6UPH).

assay experiment (Fig. 2a, b and Supplementary Fig. 2a, b). We tried to use the cryo-EM method to determine the structure of the CENP-A$^{Cse4}$ CEN3 DNA nucleosome in complex with the CBF3core. However, the CEN3 DNA again dissociated from the nucleosome bound to CBF3core on the cryo-EM grid, and the antibody fragment could not bind to the CBF3core–nucleosome complex (Supplementary Fig. 2b).

To stabilize the nucleosome, we engineered a hybrid DNA, CEN3-601, by using the CDEIII and its neighboring regions to substitute the corresponding region in the 601 DNA (Fig. 2a and Supplementary Fig. 2c). We found only the CBF3core was associated with the nucleosome when CBF3 was mixed with the nucleosome; Ndc10 dissociated from the complex (Fig. 2b). CBF3core showed a similar affinity to the CENP-A$^{Cse4}$ nucleosome with either the native CEN3 or the CEN3-601 DNA (Supplementary Fig. 2d). CBF3 and CBF3core bound to the CEN3 nucleosome with similar affinity (Supplementary Fig. 2e, f). Also, only dephosphorylated CBF3core or the CBF3core with the L1 loop deletion mutant of Skp1 could bind to the CEN3-601 nucleosome (Supplementary Fig. 2g, h). We also found that CBF3core bound to the CENP-A$^{Cse4}$ nucleosome only slightly better than the H3 nucleosome and deletion of the L1 loop in CENP-A$^{Cse4}$, a major difference between H3 and CENP-A$^{Cse4}$ on the surface of the nucleosomes, showed little effect on binding affinity (Fig. 2c). However, a single base pair shift in the incorporated DNA position led to a weaker binding of CBF3core to the

nucleosome (Fig. 2d). We were able to observe intact particles of the CBF3core bound to the CEN3-601 H3 nucleosome and obtained the cryo-EM density map at 4.2 Å resolution. It allowed us to build the structural models that showed multiple conformations for the major core region of the CBF3core (Fig. 2e, f, Supplementary Fig. 3 and Table 1)[32,33].

In the CBF3core–nucleosome complex, CBF3core bound to the nucleosome through specific recognition of the CCG motif by the Gal4 domain in one of the two Cep3 subunits (Figs. 2f and 3a). The CDEIII$^R$ DNA region associated with H2B in the free CEN3 CENP-A$^{Cse4}$ nucleosome was detached, and the full CDEIII$^R$ DNA was in the naked form (Fig. 3a). The CDEIII$^R$ site that interacts with the L1 loop and the α1 helix of core histone H2B in the free nucleosome is ~15 bp away from the GCC motif site. This structural feature suggests that the Gal4 domain binding to the CCG motif could have an allosteric inhibitory effect on the binding of the DNA by H2B (Supplementary Fig. 4a)[47]. Notably, recent studies have also shown that the binding of the pioneer transcription factor Sox2/Sox11 HMG domain to the nucleosome can also lead to the detachment of DNA from the core histones through allosteric effects[48]. In addition, in our case, it appears that the Gal4 domain binding also makes the CDEIII$^R$ DNA region more rigid as observed by cryo-EM.

The major core region of the CBF3core showing multiple conformations is connected to the Gal4 domain through a flexible linker and moves as a rigid body (relative to the Gal4-nucleosome

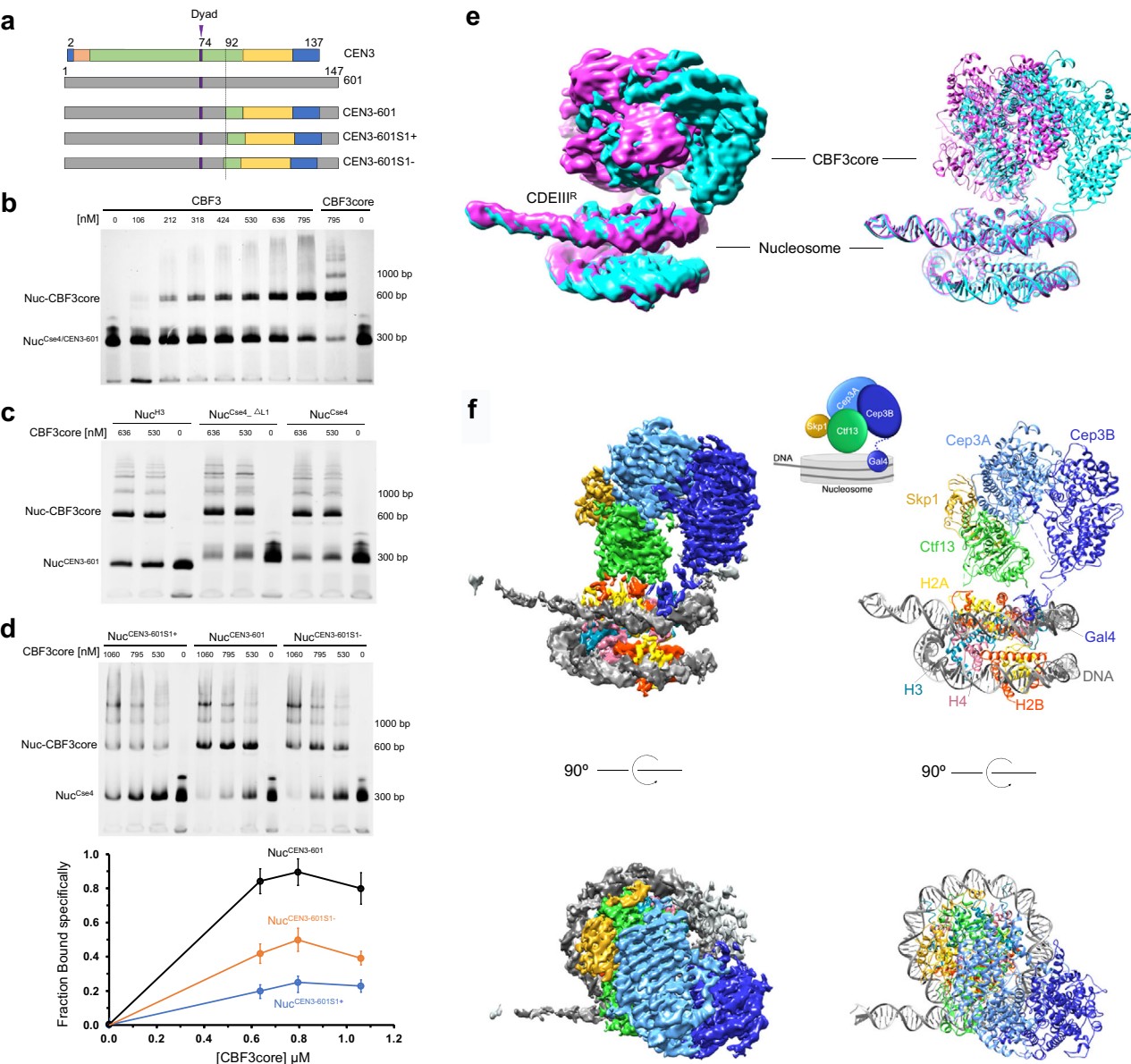

**Fig. 2 The cryo-EM structure of CBF3core bound to the CEN3-601 nucleosome. a** Illustration of engineering of the CEN3-601 DNA sequence and single bp shifts of the incorporated DNA. The alignment of the CEN3 and 601 DNA sequences are based on the structural alignment of the CEN3 CENP-A$^{Cse4}$ and human 601 CENP-A nucleosomes (PDB: 6BUZ). **b** EMSA assay of CBF3 and CBF3core binding to the CEN3-601 DNA nucleosome, showing that Ndc10 was competed out by the nucleosome and only CBF3core bound to the nucleosome. Three experiments were repeated independently with similar results. **c** EMSA assay of CBF3core binding to the CEN3-601 DNA nucleosome containing CENP-A$^{Cse4}$, CENP-A$^{Cse4}$ with deletion of the L1 loop (CENP-A$^{Cse4\_\Delta L1}$), and H3 histones. Two experiments were repeated independently with similar results. **d** EMSA assay of the effects of single bp shift of the CCG location in the CEN3-601 DNA on the binding affinity between CBF3core and the nucleosomes (top). The quantified intensity ratio of the CBF3core–nucleosome complex (first band above the nucleosome) over the total nucleosome (bottom). Data were presented as mean values. Error bars represent standard deviation values from three ($n = 3$) independently performed experiments. **e** Density maps of the CBF3core–nucleosome complex showing two extreme conformations in which the main core regions of CBF3core show different conformational movement relative the nucleosome-Gal4 region. **f** Different views of the CBF3core–nucleosome structure with CBF3core in one conformation. Middle top shows the cartoon of the CBF3core–nucleosome complex.

region) (Fig. 2e, f). Examination of the available structures of the CBF3core in the free and DNA-bound forms shows that the major core regions have the same structure but display different orientations relative to the Gal4 domain (Supplementary Fig. 4b, c)[25,32,33]. In some of the conformations, the Ctf3 subunit is close to the α2 helix of H2B and the L1 loop of H3 (Fig. 2e and Supplementary Fig. 4d). It explains why scFv could not bind to the CBF3core–nucleosome complex (Supplementary Figs. 2a and 4e).

To examine the interactions between the CBF3core and the CEN3 CENP-A$^{Cse4}$ nucleosome, we built a structural model by

substituting the H3 histone in the CEN3-601 nucleosome–CBF3core complex with CENP-A$^{Cse4}$. We found that the Cbf13 subunit was close to the five charged residues of the α2 helix of H2B and the three residues in loop 1 of CENP-A$^{Cse4}$ (Fig. 3b). The model suggests potential formation of a salt bridge between Ctf13 K282 and H2B E117 and hydrophobic interactions between Cft13 P297 and CENP-A$^{Cse4}$ K1987 (Supplemental Fig. 4d). We mutated the residues in the α2 helix of H2B to Ala, deleted the three residues in loop 1 of CENP-A$^{Cse4}$, and measured the binding affinities of CBF3core to the

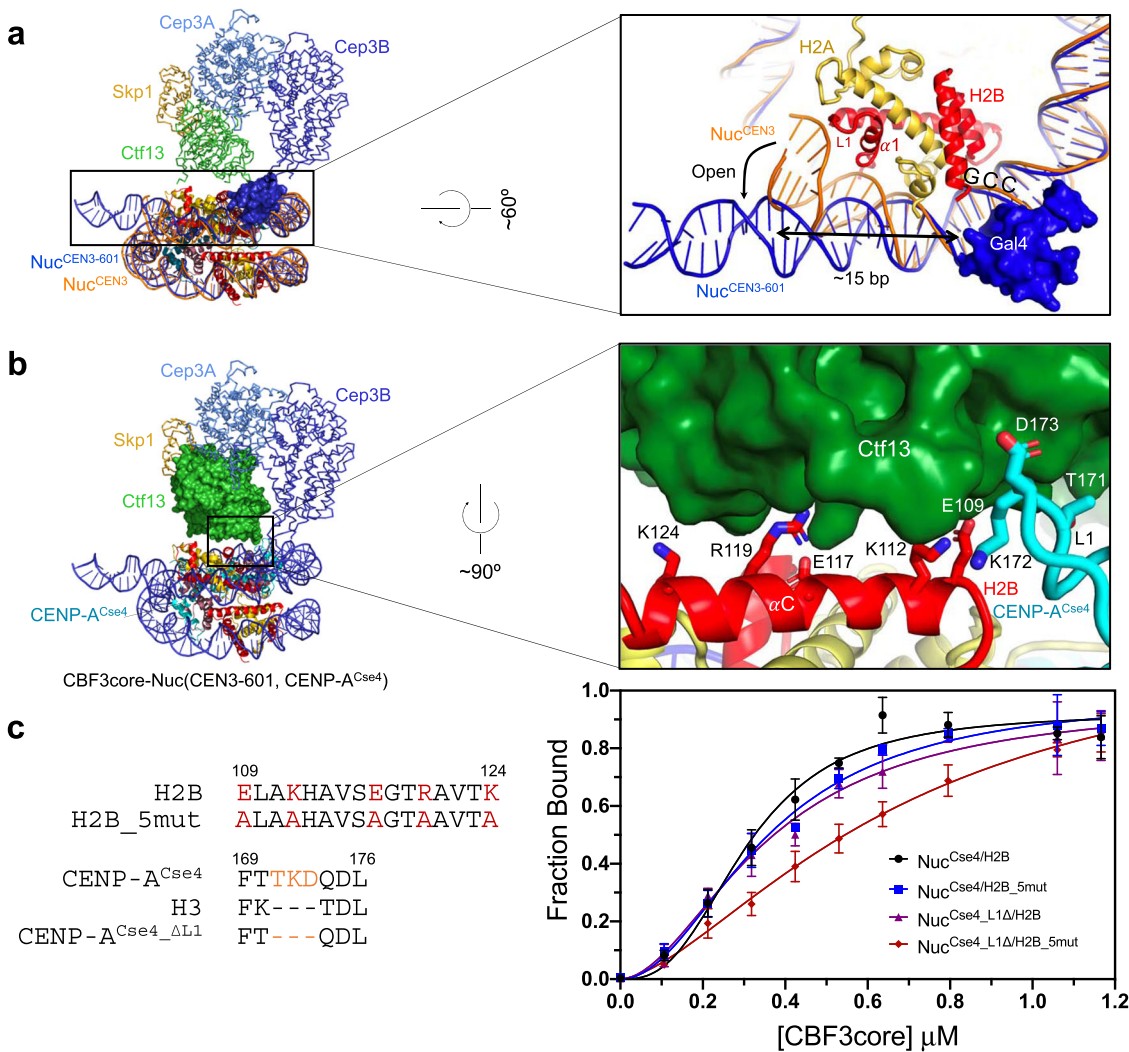

**Fig. 3 Interactions between CBF3core and the nucleosome. a** Allosteric effects of Gal4 domain binding on DNA. The structures of the CBF3core–nucleosome complex and the free CEN3 nucleosome were aligned on the core histones. DNA in the CBF3core–nucleosome complex is colored blue and DNA in CEN3 nucleosome is colored orange. Gal4 domain is shown in blue surface. **b** A structural model for interactions between the core histones and CBF3core in one conformation. H3 in the CBF3core–CEN3-601 nucleosome structure was substituted by CENP-A$^{Cse4}$ in the CEN3 CENP-A$^{Cse4}$ nucleosome though alignment of the backbones of core histones. In the model, the residues in the α2 helix of H2B histone and loop 1 of CENP-A$^{Cse4}$ are close to the Ctf13 subunit of CBF3core (green surface). **c** Measurement of the effects of histone mutations on the binding affinity between CBF3core and the CEN3 CENP-A$^{Cse4}$ nucleosome. Mutant H2B_5mut had the five residues (Glu109, Lys112, Glu117, Arg119, and Lys124 in sticks) in the α2 helix of H2B (red) were mutated to Alanine. Mutant CENP-A$^{Cse4}$_$_{ΔL1}$ had three residues (Thr171, Lys172, and Asp173 in sticks) in the L1 loop of CENP-A$^{Cse4}$ (cyan) deleted. Nitrogen and oxygen atoms for the mutated residues are shown in blue and red colors, respectively. The $K_d$ of CBF3core binding to Nuc$^{Cse4/H2B}$, Nuc$^{Cse4/H2B\_5mut}$, Nuc$^{Cse4\_ΔL1/H2B}$, Nuc$^{Cse4\_ΔL1/H2B\_5mut}$ are 0.32 ± 0.06 μM, 0.35 ± 0.05 μM, 0.36 ± 0.05 μM, and 0.68 ± 0.04 μM. Data were presented as mean values. Error bars represent standard deviation values from three ($n = 3$) independently performed experiments. Source data are provided in the Source Data file.

nucleosomes containing the histone mutants (Fig. 3c and Supplementary Fig. 4f). The binding dissociation constant ($K_d$) for the nucleosome with wild type histones is ~0.32 μM. Each of the two mutants showed little effects on the binding affinity, and when combined, they decreased the binding affinity by only less than a factor of two (Fig. 3c). These results showed that CBF3core only made weak and dynamic contacts with the core histones.

## Discussion

In this study, we determined the structure of the CEN3 CENP-A$^{Cse4}$ nucleosome, which shows that the CEN3 DNA mainly uses CEDII and CDEIII to interact with the core histones. In the structure, the CDEI is in the linker DNA region, allowing for binding by CBF1 in a specific sequence-dependent manner. Our structure is consistent with the in vivo nucleosome mapping results, which show that no significant region beyond CEN3 CDEs are resistant to MNase digestion[42,43]. In contrast, in the recently proposed "proximity" model for CENP-A$^{Cse4}$ deposition by CBF3, the CDEIII is located in the linker DNA region[25]. Also, the dyad position in the CEN3 CENP-A$^{Cse4}$ nucleosome structural model that is proposed based on hydroxyl radical footprinting results and computational modeling is ~12 nucleotide away from the dyad in our structure[24], leading to the conclusion that the Mif2 AT-hook domain contacts only one side of the nucleosome dyad[44]. Using the dyad location in our cryo-EM structure (position at 74) (Fig. 1), Mif2 would contact both sides of the nucleosome dyad. These results show our understanding of the determinant of nucleosome positioning is limited and it is still

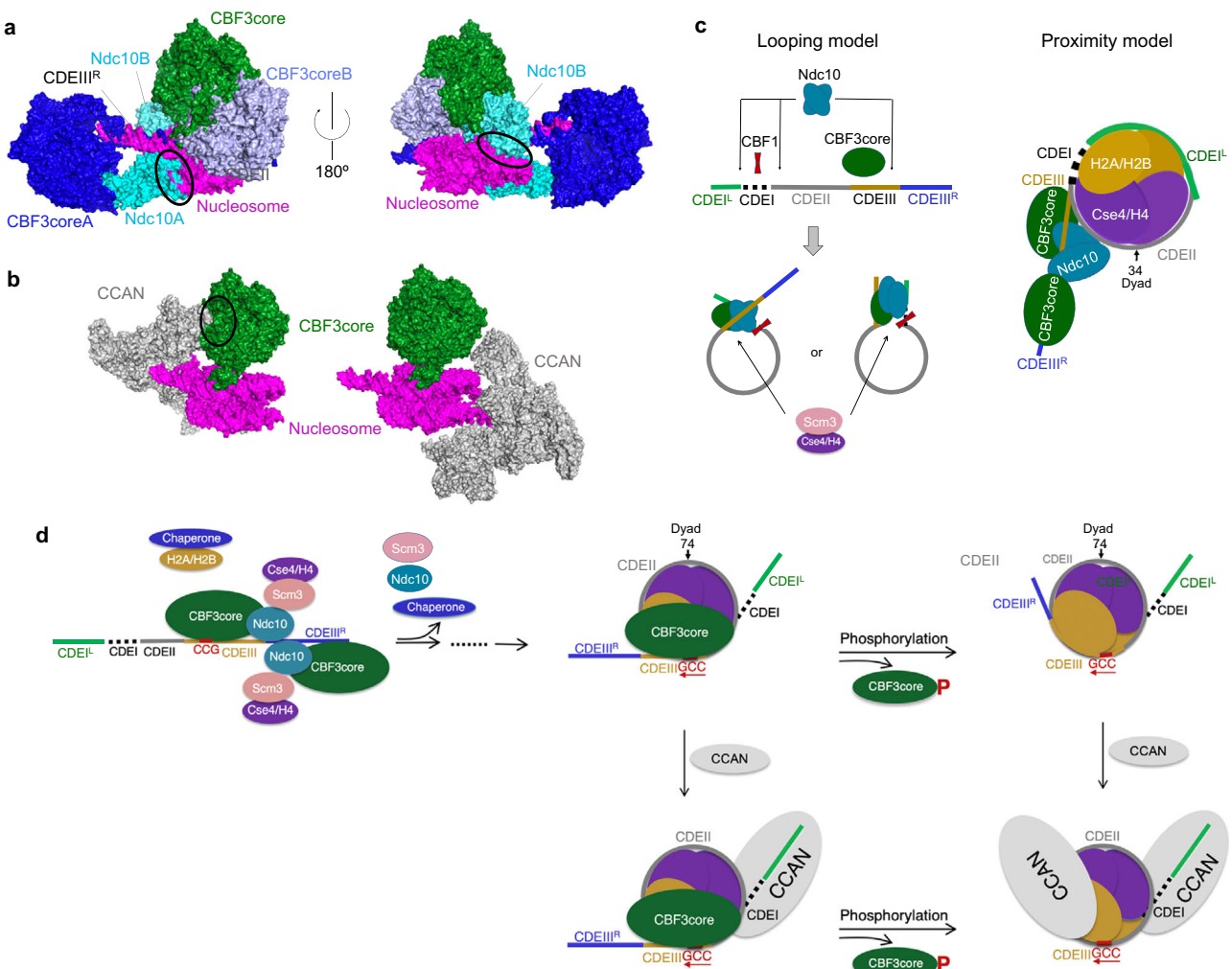

**Fig. 4 Proposed mechanism of CBF3-guided formation of the CENP-A$^{Cse4}$ CEN3 nucleosome and its association with CCAN. a** Overlay of the structures of the CBF3core–nucleosome and (CBF3)$_2$-DNA (PDB ID: 6GYS) complexes through alignment of the CCGs region and Gal4 domains in the two structures. CBF3core binds to DNA and the nucleosome in different orientations and the two Ndc10 subunits (cyan) have steric clash with the nucleosome (magenta) as indicated by circles (black). **b** Overlay of the structures of CBF3core–nucleosome and CCAN–nucleosome complexes (PDB ID: 6Q1D) by aligning the nucleosome structures. CCAN bonding to the CDEIII$^R$ DNA (left) can cause steric clash with the CBF3core as indicated by the circle (black), whereas its binding to the CDEI DNA side can be accommodated by the binding of CBF3core (right). **c** Diagram illustration of "looping" (left) and "proximity" (right) models. The dyad in the proximity model is at nucleotide sequence number 34 (Fig. 1a). **d** Diagram illustration of the speculated structural and dynamic mechanism. After DNA replication, the CEN3 DNA is recognized by the CBF3 dimer. Ndc10 in the CBF3 associates with Scm3 bound to CENP-A$^{Cse4}$-H4. The chaperones of H2A–H2B and CENP-A$^{Cse4}$-H4 deliver the histones to form the CENP-A$^{Cse4}$ nucleosome–CBF3core complex, which could involve various intermediate states (dashed line between the arrows). Note that the dissociated Ndc10–Scm3 might still interact with the linker or nucleosomal DNA in a DNA sequence-independent manner. Also, CDEI could be available for binding of CBF1 in both the CBF3core–CEN3 nucleosome complex and free CEN3 nucleosome. The CBF3core–CEN3 nucleosome complex can only recruit one copy of CCAN on the CDEI side. The free CEN3 nucleosome can recruit two copies of CCAN.

not possible to predict nucleosome positioning accurately. In particular, it is intriguing that the CEN3 nucleosome is uniquely positioned considering the fact that its sequence is highly AT-rich.

We also find that CBF3 can bind to the CEN3 CENP-A$^{Cse4}$ nucleosome while causing dissociation of Ndc10 from the CBF3core–nucleosome complex. Alignment of the structures of the CBF3core–nucleosome$^{CEN3-601}$ and (CBF3)$_2$–DNA complex[25] on the Gal4 domain reveals steric clashes of Ndc10 with the nucleosome core (Fig. 4a), which explains the dissociation of Ndc10 from the CBF3core bound to the nucleosome$^{CEN3-601}$ and Ndc10 binds to CBF3core weakly[32]. The alignment also shows a substantial difference in the orientation of the major core region of the CBF3core in the two structures. Also, the structural alignment

of the CBF3core–nucleosome and the constitutive centromere-associated network (CCAN) or inner kinetochore complex bound to the nucleosome[49] shows that one copy of CCAN can co-bind to the nucleosome with CBF3core (Fig. 4b).

Based on the above results, we speculate a structural and dynamic mechanism for a two-step process for specifying the budding yeast centromeric nucleosome (Fig. 4c, d)[50]. After DNA replication, CBF3, in association with the Scm3–Cse4–H4, could target centromeric DNA by specifically recognizing the CCG motif in CDEIII[25]. Upon the CENP-A$^{Cse4}$ deposition to the CEN DNA by Scm3, the CENP-A$^{Cse4}$ nucleosome could partially form. Meanwhile, the CBF3core is still attached to the CDEIII DNA. During this process, intermediate steps involving various partially formed nucleosomes might occur[29] and Scm3–Ndc10 dissociates

from the CBF3core but could still interact with the linker or nucleosomal DNA in a sequence-independent manner[28,36]. In our model, Ndc10 would serve as a dynamic adapter between CBF3core and Scm3, instead of playing an architectural role in the previous "looping" and "proximity" models[25,36]. The dynamic interactions of CBF3core with the core histones could help prevent the core histones from dissociating from the DNA, compensating the weak affinity between the core histones and the AT-rich DNA. The AT-rich sequence in CEN3 could also play a role in enhancing the binding specificity of the CCG motif by the Gal4 domain. The position of the CEN3 in our cryo-EM structure allows CBF3core to bind the nucleosome while Ndc10B would clash with the nucleosomal DNA. Notably, in the "proximity" model, Ndc10B can coexist with the nucleosome. However, the nucleosome with the hypothetical dyad at nucleotide position 34 is not at an intrinsically favored position. It might only exist as a transient intermediate. The CEN3 CENP-A$^{Cse4}$ nucleosome–CBF3core complex is capable of co-binding of one copy of CCAN. The specific recognition of CENP-A$^{Cse4}$ by CCAN was achieved by the CENP-C motif in its subunit Mif2[44,49,51]. Upon phosphorylation, CBF3 could dissociate from the complex, and the fully structured nucleosome could form, which allow binding of two copies of CCAN.

We want to emphasize that the events described in these models are based on in vitro structural studies. Whether they occur in vivo remain to be tested. In addition, inside the cell, these structural events are likely regulated to control the binding of DNA at the same site by different proteins, and the reversal process might also occur. Results from available in vivo studies are consistent with some of the structural events described in our model. For example, the inner kinetochore proteins and CBF3 can coexist on the Cse4 nucleosome, which is coordinated by the Ctf19–Mcm21–Okp1 complex[52,53]. Also, consistent with the dissociation of Ndc10 after the formation of the centromeric nucleosome, Ndc10 is enriched at the spindle midzone in late anaphase[54].

Finally, the DNA sequence-dependent deposition of CENP-A$^{Cse4}$ facilitated by CBF3 and Scm3 has implications for understanding the analogous DNA sequence-dependent histone deposition in human centromeric nucleosomes. Like CBF3, human CENP-B recognizes the 17 bp CENP-B box DNA motif in every other α-satellite DNA and is associated with the centromere protein CENP-C[55]. CENP-C binds to HJURP[56], a specific chaperone of human CENP-A for its deposition[57,58]. Similarly, CENP-B may also associate with Daxx, a specific chaperone of histone variant H3.3, and facilitate its deposition at the centromere DNA[59]. The DNA sequence-dependent function of CENP-B is important for enhancing chromosome segregation fidelity[60] and for generating the artificial human chromosomes[61]. These results underscore the functional role of DNA sequence in histone deposition and nucleosome formation at the centromere. However, it should be noted that although the above DNA sequence-dependent centromeric nucleosome formation shows an analogous molecular mechanism, for cell and organismal viability, CBF3 is essential, whereas CENP-B is dispensable[62]. It suggests that epigenetic factors play a more important role in human centromere function.

## Methods

**Expression and purification of histones**. Recombinant histones H3, H4, CENP-A$^{Cse4}$, and CENP-A$^{Cse4}$_△L1 (generated using QuikChange kit, Agilent; the primers are listed in Supplementary Table 2) were expressed individually in *Escherichia coli* BL21(DE3) cells and purified using established protocols[63]. *E. coli* cells harboring each histone expression plasmid were grown at 37 °C in 2 x YTB Broth. When OD$_{600}$ reached around 0.6–0.8, 0.3 mM IPTG added to induce recombinant protein expression for 3 h at 37 °C. The cells were harvested and resuspended in 50 ml of buffer A (50 mM Tris-HCl, 500 mM NaCl, 1 mM PMSF, 5% glycerol, pH

8.0), followed by sonication on ice for 360 s with a pulse of 15 s on and 30 s off. The cell lysates were centrifuged at 140,000 × g for 20 min at 4 °C. The supernatant was discarded. The pellet containing histones was resuspended in 50 ml of buffer A and 7 M guanidine hydrochloride. The samples were rotated for 12 h at 4 °C, and the supernatant was recovered by centrifugation at 140,000 g for 60 min at 4 °C. The supernatants were dialyzed against buffer C (5 mM Tris-HCl, pH 7.4, 2 mM 2-mercaptoethanol, 7 M urea) for three times. The supernatant was loaded to Hitrap S column chromatography (GE Healthcare). The column was washed with buffer D (20 mM sodium acetate, pH 5.2, 200 mM NaCl, 5 mM 2-mercaptoethanol, 1 mM EDTA, and 6 M urea). The histone protein was eluted by a linear gradient of 200 to 800 mM NaCl in buffer D. The purified histones were dialyzed against water for three times, and freeze-dried.

Recombinant H2A–H2B and H2A–H2B_5mut dimers (generated by QuikChange kit, Agilent; the primers are listed in Supplementary Table 2) were expressed in *E. coli* JM109(DE3) cells and purified as described[28]. The purification protocol was similar to the one described above except that the pellet containing H2A–H2B dimers was resuspended in 25 ml of buffer A containing 0.2 M HCl. The samples were frozen for 30 min at −20 °C. After melting, centrifugation was applied to the samples. Eight milliliters of 2 M Tris was added to the supernatant. The proteins were purified using the Hitrap S column chromatography, and the purified H2A–H2B dimers were collected and stored at −80 °C.

**Overexpression and purification of CBF3**. The N terminal domain of Ndc10 (NTD,1-544) and CBF3core containing Cep3 dimer, Skp1 or Skp1_ΔL (37–64 deleted), and Ctf13 were purified using established protocols[32]. To obtain the dephosphorylated CBF3core, the NEBuffer for protein metallophosphatases (PMP) and MnCl$_2$ were added into the sample, then followed by lambda protein phosphatase. The mixture was incubated at 30 °C for 30 min. The dephosphorylated CBF3core was purified by size-exclusion chromatography (Superose 6, 10/100, GE Healthcare) in the final buffer of 10 mM HEPES 7.3, 300 mM NaCl, and 1 mM DTT. The purified proteins were collected and stored at −8 0 °C. To obtain the dephosphorylated CBF3, purified dephosphorylated CBF3core and Ndc10 NTD were mixed at 1:1 molar ratio and incubated at 4 °C for 2 h. The mixture was loaded onto Superose 6 Increase 10/100 (GE Healthcare). The peak fraction was assessed by SDS–PAGE and Coomassie stain. The dephosphorylated CBF3 was stored at −80 °C.

**Preparation of DNA**. The 136 bp CEN3 DNA was prepared as described[44]. The 147 bp CEN3-601 DNA, which contains 48 bp DNA of CBF binding site underlined, were prepared by PCR amplification followed by ethanol precipitation and purified using the POROS column.
The forward and reverse template DNA sequence are: ATCGAGAATCCCGGTGCCGAGGCCGCTCAATTGGTCGTAGACAGCTC-TAGCACCGCTTAAACGCACGTACGCGCTGTCCCCCGCGTTTT and ATCGGATGA̲T̲T̲T̲C̲T̲T̲A̲C̲T̲A̲T̲T̲T̲C̲T̲T̲T̲T̲T̲T̲A̲A̲C̲T̲T̲T̲C̲G̲G̲A̲A̲A̲T̲C̲A̲A̲A̲T̲A̲-CACTAATA̲TTAAAACGCGGGGGACAGCGCGTACGTGCGT, respectively. The forward and reverse primer sequences are: ATCGAGAATCCCGGTG and ATCGGATGATTTCTTACTATTTC, respectively (Supplementary Table 2).
The PCR products were pelleted using 70% ethanol containing 0.3 M NaAc 5.2. The sample was incubated for 40 min at −20 °C, followed by centrifugation. The supernatants were discarded, and the pellet was washed by 70% ethanol twice, then resuspended by TE buffer. The sample was loaded to POROS column chromatography (GE Healthcare). The column was washed with buffer E (20 mM Tris-HCl, pH 7.4, 5 mM 2-mercaptoethanol), and the DNA was eluted by a linear gradient of 0 to 1 M NaCl in buffer E. The purified DNA were collected and stored at 4 °C. All described hybrid DNAs were prepared and purified with the same procedure.

**Reconstitution of nucleosomes**. Core histone octamers were reconstituted first as described[12]. Purified recombinant histones in equal stoichiometric ratio were dissolved in unfolding buffer (7 M guanidine-HCl, 20 mM Tris-Cl at pH 7.4, 10 mM DTT). The mixtures were dialyzed against refolding buffer (10 mM Tris-Cl at pH 7.4, 1 mM EDTA, 5 mM β-mercaptoethanol, 2 M NaCl, 0.1 mM PMSF) for 1 day at 4 °C twice. The mixture was centrifuged at 3500×g to remove any insoluble material. Soluble octamers were purified by size fractionation on a Superdex 200 gel filtration column.
All yeast CEN3 nucleosomes were reconstituted following the published protocol[12]. Briefly, purified histone octamers and 136 bp CEN3 DNA were mixed with a 1:1.3 ratio of DNA:octamer in high-salt buffer (2 M NaCl, 10 mM K/Na-Phosphate at pH 7.4, 1 mM EDTA, 0.02% NP-40, 5 mM β-mercaptoethanol). The 1 ml mixture in a dialysis bag was placed in 600 mL of the high-salt buffer and dialyzed for 60 min followed by salt gradient dialysis. Four liters of a low-salt buffer (100 mM NaCl, 10 mM K/Na-Phosphate at pH 7.4, 1 mM EDTA, 0.02% NP-40, 2 mM β-mercaptoethanol) were gradually pumped into dialysis buffer with a flow rate of 2 ml/min for 30 h. The dialysis bag was then dialyzed against low-salt buffer for 60 min. The dialysis was done at room temperature. The sample was then incubated at 65 °C for 12 h. The mixture was centrifuged at 12,000×g to remove any insoluble material. The soluble nucleosomes were stored at 4 °C for less than 1 week.

All yeast CEN3-601, CEN3-601S1 + ,and CEN3-601S1⁻ nucleosomes were reconstituted using the same protocol, except that the samples were incubated at 37 °C for 2 h. Then nucleosomes were further purified by ion-exchange chromatography (TSKgel DEAE, TOSOH Bioscience, Japan) to remove free DNA and histones. The purified nucleosomes were dialyzed against TE buffer containing 10 mM Tris 7.4, 1 mM EDTA, and 2 mM DTT twice.

**Preparation of the complex of the CEN3 CENP-A^Cse4 nucleosome bound to scFv.** We found that the solubility of the CEN3 CENP-A^Cse4 nucleosome was sensitive to the temperature. The soluble nucleosomes were precipitated by incubating at 4 °C for over 12 h. The sample was centrifuged at 12,000 × g for 1 min to remove the supernatants. The pellet was resuspended in TE buffer. scFv was mixed with the CEN3 CENP-A^Cse4 with a 3:1 ratio of scFv:nucleosome at room temperature for 1 h. Then the samples were concentrated for electron microscopy analyses.

**Preparation of CBF3core complex with the CEN3-601 nucleosome.** The CBF3core and CEN3-601 nucleosome complex was reconstituted by mixing purified dephosphorylated CBF3core with the CEN3-601 nucleosome at 4 °C for 1 h. The mixed sample was dialyzed for overnight in a buffer of 10 mM HEPES at pH 7.3, 50 mM NaCl, 1 mM EDTA, and 1 mM DTT at 4 °C. The complex was purified and stabilized using the GraFix method as described[64]. Briefly, the top solution contained 10 mM HEPES at pH 7.3, 50 mM NaCl, and 10% glycerol (Sigma). The bottom solution contained 10 mM HEPES a pH 7.3, 50 mM NaCl, 0.15% glutaraldehyde (Polysciences), and 30% glycerol (Sigma). After ultracentrifugation at 190,000 × g for 18 h, the best fraction was collected and dialyzed against the buffer containing 10 mM Tris-HCl at pH 7.4, 50 mM NaCl, and 2 mM DTT. Then the samples were concentrated for electron microscopy analyses.

**Cryo-EM sample preparation and data collection.** Three microliters of nucleosome–scFv sample was loaded onto a glow-discharged holey carbon grid (Quantifoil 300 mesh Cu R1.2/1.3), and 3 μL of nucleosome–CBF3core sample was loaded onto a glow-discharged Lacey grids. The grids were blotted for 3 s at 20 °C and 100% relative humidity using a FEI Vitrobot Mark IV plunger before being plunge-frozen in liquid nitrogen-cooled liquid ethane. The samples were first screened a 200 kV microscope Tecnai F20. Final Cryo-EM datasets were collected using a Titan Krios G3 electron microscope (Thermo-Fisher) operated at 300 kV. Micrographs were acquired at the nominal magnification of 130,000x (calibrated pixel size of 1.06 Å on the sample level) using a Gatan K2 Summit direct electron detection camera equipped with a Gatan Quantum LS imaging energy filter with the slit width set at 20 eV. The dose rate on the camera was set to 8 e⁻/pixel/s. The total exposure time of each micrograph was 10 s fractionated into 50 frames with 0.2 s exposure time for each frame. The data collection was automated using the Leginon software package[65].

**Image processing.** The data processing procedures were shown in supplementary Fig. 1 for nucleosome–scFv dataset and supplementary Fig. 3 for nucleosome–CBF3core dataset. The nucleosome–scFv dataset was processed using RELION/3.0-beta2 and the nucleosome-CBFcore dataset was processed using RELION/3.0.7 following the standard procedures in RELION3[66]. The beam-induced image drift was corrected using MotionCor2[67]. The averaged images without dose weighting were used for defocus determination using CTFFIND4[68] and images with dose weighting were used for particle picking and extraction. Particles were picked by Gautomatch (https://www.mrc-lmb.cam.ac.uk/kzhang/Gautomatch/) using templates generated from datasets collected on a 200 kV microscope Tecnai F20.

For the scFv-nucleosome dataset, 342,749 particles were picked. Bad particles were removed by 2D classification. Then 192,020 particles were selected from 3D classification with two classes with good structural features. After re-centering, the best 143,164 particles were selected for consensus refinement. After Bayesian polishing, 3D auto-refine and post processing, a 3.1 Å map was generated for model building.

For nucleosome–CBF3core dataset, 827,118 particles were picked. Bad particles and free nucleosome particles were removed in 2D classification. 194,679 particles were applied for 3D classification, and free nucleosome particles were further discarded. After re-centering, 115,666 particles were selected for 3D refinement. The blurry density in 2D class averages and 3D reconstruction suggest the flexibility between nucleosome and CBF3core. We divided the particles into nucleosome part and CBF3core part using density subtraction. 3D auto-refine and post processing generate the final 4.0 Å map of density subtracted CBF3core and 4.2 Å map of density subtracted nucleosome. To elucidate the flexibility of the complex, multibody refinement was applied to study the relative motion of CBF3core to nucleosome (Supplementary Movie 1).

**Model building and structure analysis.** For the scFv–CEN3 nucleosome complex, an initial model of the CENP-A nucleosome histone octamer and scFv was generated using the free CENP-A nucleosome structure reconstituted with human histone proteins (PDB: 6O1D)[18]. The model was fitted into the cryo-EM density map of scFv–CEN3 nucleosome complex. The CEN3 DNA sequence was built into

the map from scratch in COOT[69] and the histone octamer and scFv were optimized by manual rebuilding. The whole complex was refined using real space refinement in PHENIX[70].

For the nucleosome–CBF3core complex, initial model of nucleosome was generated with a rigid body fit into the density using previously built scFv–CEN3 nucleosome structure. DNA sequence was changed to CEN3-Widom 601 sequence based on CENP-A nucleosome with a Widom 601 DNA structure (PDB: 6BUZ)[13]. Initial model was generated by a rigid body fitting using CBF3–CEN3 complex structure (PDB: 6GYS). Nucleosome and CBF3core structures were optimized by manually rebuilding in COOT followed by further refinement using real space refinement in PHENIX. Figures were made using UCSF Chimera[71] and PyMOL (Version 1.8, Schrödinger, LLC. DeLano Scientific).

**Electrophoretic mobility shift assay.** Typical binding reactions of complex formation between CBF3/CBF3core and CEN3-601 nucleosomes were carried out for 60 min on ice in 10 mM Tris at pH 7.4, 75 mM NaCl, and 1 mM DTT. Reactions contained 530 nM nucleosome, and either 106, 212, 318, 424, 530, 636, 795, 1060, and 1166 nM CBF3, or CBF3core. Ten microliters of the binding reactions were analyzed on 4% acrylamide gels in 0.2 x TBE 100 V for 90 min at 4 °C. No ethidium bromide (EtBr) was added at this point to prevent potential disruption of DNA structure by EtBr. After electrophoresis, gels were stained with EtBr and the band intensity was quantified using Image J. The first band above the free nucleosome was taken as the CBF3core–nucleosome complex assuming 1:1 ratio between CBF3core and the nucleosome. The fraction bound is calculated as the ratio of the intensity between this band and that of total free nucleosome. Binding data were fitted with the Hill equation and analyzed in Prism (Graphpad). Binding of CBF3core does not affect EtBr staining of the nucleosome.

For the binding reactions between CBF3/CBF3core and CEN3 nucleosomes, the mixture was incubated at room temperature for 60 min. Ten microliters of the binding reactions were analyzed by electrophoresis at 100 V for 20–30 min on native agarose gels (Seakem ME and Lonza LE) in 0.2 x TBE. After electrophoresis, gels were stained with SYBR Green I (Invitrogen) and visualized with a Fujifilm LAS-3000 camera. Images were exported into TIFF files for quantification using Image Quant software (Amersham Biosciences).

**Reporting Summary.** Further information on research design is available in the Nature Research Reporting Summary linked to this article.

## Data availability

Three-dimensional cryo-EM density maps have been deposited in the Electron Microscopy Data Bank under accession numbers EMDB-22696 (CEN3 CENP-A^Cse4 Nucleosome–scFv), EMDB-22698 (CEN3-601 H3 Nucleosome in Nucleosome–CBF3core), and EMDB-22697 (CBF3core in Nucleosome–CBF3core). The coordinates of atomic models have been deposited in the Protein Data Bank under accession numbers 7K78 (CEN3 CENP-A^Cse4 Nucleosome–scFv), 7K7G (CEN3-601 H3 Nucleosome in Nucleosome–CBF3core), and 7K79 (CBF3core in Nucleosome–CBF3core). Previously published structures used in this study: 6O1D, 6BUZ, 6GYS, 6UPH, 6FE8, 6F07, and 6Q1D are available in PDB databank. All other relevant data supporting the key findings of this study are available within the article and its Supplementary Information files or from the corresponding authors upon reasonable request. Source data are provided with this paper.

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

## Acknowledgements

We are grateful to Dr. Jiansen Jiang for the cryo-EM resource and guidance. We thank Dr. Hua Xiao for plasmids and suggestions on reconstitution of the CEN3 nucleosome, and Drs. Alex Kelly, Munira Basrai, Prashant Mishra, and Kentaro Ohkuni for helpful discussion. EM datasets were collected on the microscope in the NIH Multi-Institute Cryo-EM Facility (MICEF) and EM data processing was carried out using the NIH HPC Biowulf cluster (http://hpc.nih.gov). We thank Dr. Huaibin Wang for technical support on electron microscope. Our work was supported by the Francis Crick Institute, which receives its core funding from Cancer Research UK (FC001155), the Medical Research Council (FC001155) and the Wellcome Trust (FC001155) (M.R.S.), and by the intramural research program of Center for Cancer Research, National Cancer Institute, National Institutes of Health (Y.B.).

## Author contributions

Y.B. and C.W. initiated the study on the CEN3 nucleosome. R.G., Y.B., and M.S. initiated the study on the CBF3–nucleosome complex. M.S. provided the CBF3 protein. B.-R.Z. provided the scFv protein. R.G. prepared the samples and conducted the biochemical experiments. E.H. assisted with mutation studies. R.G. and T.L. collected and processed the cryo-EM data. R.G. built the structural models. R.G. and Y.B. analyzed the structures and wrote the paper with input from all other authors.

## Funding

## Competing interests

The authors declare no competing interests.
