## [Peer Review File · Nature Communications]

Reviewers' Comments:

Reviewer #1:

Remarks to the Author:

The budding yeast centromere has been a powerful model for understanding eukaryotic centromeres for 40+ years, and is in the midst of important new structural insight from a handful of labs. Budding yeast is exceptional because DNA sequence specifies location of the centromere, unlike most eukaryotes where centromeres can exist on diverse DNA sequences. In this study, the authors solved structures of yeast CENP-ACSE4 nucleosomes on CEN3 DNA (bound by stabilizing antibody fragments) and CENP-ACSE4 nucleosomes on hybrid 601-CEN3 DNA bound by CBF3 (essential sequence specific binding complex). They also use biochemical approaches to address the role of CBF3 binding to CCG region on CEN3/601 DNA on CENP-A nucleosomes and oppose a model put forward of 'proximity looping' by the Barford lab in 2018. In the present study, the authors report selectivity of CBF3 core (without NDC10) over the full CBF3 complex and propose another mechanism for CBF3 interaction on nucleosomes. The authors also report structural details of CBF3 interactions on the centromeric nucleosome. Further, the authors propose the role/ mechanism of yeast genetic component CBF3 might act analogous to human genetic factor CENP-B in the CENP-A assembly and model of yeast CCAN assembly on CBF3 bound CENP-A nucleosomes. Overall the study has the potential to be of high interest throughout the centromere community, providing structural data that broadly addresses the role of inter-subunit interactions for CBF3 and CENP-A nucleosomes. I list several major and minor concerns that should be remedied before a revised version is considered for publication.

Major concerns:

1. The presentation of the position of the CENP-A nucleosome on CEN DNA is specifically confusing (it is also generally an insufficiently scholarly discussion; see point #2, below). The authors cite the positioning data coming from footprinting data in vivo (Xiao et al GD 2017 (ref 27)). But the sequence they use is quite shifted (by 12 nt if I am reading it correctly). The authors should explain any reason for these differences in the results section. Are there implications from their structure in relation to the 2017 paper regarding the sequence elements that Mif2 recognizes for its marked increased affinity for nucleosomes wrapped with CEN DNA? Does the dyad position impact on their model of CBF3-directed CENP-A nucleosome assembly?
2. Nothing is cited for the central issue of mapping CENP-A nucleosome sequences in any species or for using those native sequences for understanding the role of DNA sequence for the structure and function of centromeric chromatin (except for one paper from this group). The native DNA sequence used here is a primary advance here relative to other yeast structural studies that primarily resort to the artificial 601 sequence, so the more important contributions from others in the field at large to this part of centromere studies really do need to be acknowledged in order for the reader to put the present findings in proper context with the published work.
3. What is implication for the need in this study to stabilize CENP-A nucleosomes by either an antibody fragment or 601 sequences introduced? Complementary experiments to understand what these artificial features 'do' to the sampling of different conformations/ states would be best. At a bare minimum, there should be some discussion what the need for these artificial stabilizing measures implies (as well as a discussion of the root cause of the instability by the extremely AT-rich CEN DNA sequence).
4. The authors mention on page 5 that core histones of CEN3 and 601 wrapped CENP-Acse4 nucleosomes have similar structures, but it is not clear the degree to which this has been measured. When aligning some landmark features of the nucleosome, what are the structural differences in these two structures? RMSDs?
5. If CEN3 DNA is used in assay shown in Fig. 2B, is there similar binding with and without NDC10?
6. How robust is Fig. 2d regarding the 1 bp affects implied by the gel shown? Quantitation is not shown (as well as for some other of the gel-based experiments), so it is challenging to assess how much faith one has in this potentially striking result.
7. How does the Gal4 position changes in the two conformations shown in Figure 2E? Can the authors

report RMSD values? Does Gal4 bind to CCG region on CENP-A nucleosome DNA in both conformations?

8. For Fig 3C, quantitation is shown, but there are very few details about this in the supplement (more details on this assay and the quantitation are needed). I infer that EtBr is used to assess bound/unbound, but DNA-binding proteins (and indeed nucleosomes) can potentially restrict EtBr intercalation. Isn't there a better way to do this?

9. It would be clearer if the authors explicitly pointed out the differences between looping and proximity model along with their model shown in Figure 4a or 4c. Can the authors explain looping and proximity model in a similar manner as the cartoons are drawn in 4c? In the left panel, I would suggest that the authors show proximity models with dyad positions (along with their model) and indicate significant differences that would affect CBF3 mediated assembly of Cse4 during DNA replications in their structural model. Also, I was confused by the apparent symmetry of binding of the CCAN in their diagram. I thought that Xiao et al 2017 reported that the two copies of the CCAN component, CENP-C/Mif2, are bound asymmetrically with respect to the CEN DNA wrapped around the CENP-A nucleosome.

10. Please add a reference for a study that demonstrates that the CCAN and CBF3 are bound to CSE4 nucleosomes simultaneously.

11. On page 10, the following sentence is very confusing: "The CEN3 CENP-ACse4 nucleosome-CBF3core complex is capable of co-binding of one copy of CCAN through the specific recognition of CENP-ACse4 by its subunit Mif227,32 in a way similar to the recognition of human CENP-A nucleosome by the CENP-A protein36." The human CENP-A nucleosome can bind two copies of CENP-C. Is the point here about the stoichiometry of components? If so, it isn't correct then. If not, then the meaning of the sentence and the reasoning for citing the human arrangement is unclear.

12. The idea to have a closing discussion on the relation of their findings involving CBF3 to what happens in species (like humans) with CENP-B could be potentially useful. But exclusively discussing the similarities, as the authors do here, is misleading. Yes, CENP-B provides some centromere function, but unlike CBF3 (or other key molecules like CENP-A/Cse4 and CENP-C/Mif2) it is dispensable for cell and organismal viability. Also, it is true that the initial types of human artificial chromosomes required CENP-B, but the latest generation of human artificial chromosomes completely bypass the need for CENP-B by seeding centromeric nucleosomes. To my knowledge, nobody is considering a scenario where CBF3 is bypassed for budding yeast centromere identity and/or function. That all said, while CBF3/CENP-B is not a perfect comparison, a somewhat extended discussion on this to close the paper would be very helpful if both the key similarities and differences are included.

Minor concerns:

13. Cite Supplementary figure 2a in Paragraph 1 in page 6.

14. Supplementary figures 2e, f are not referenced in the text.

Reviewer #2:

Remarks to the Author:

The authors pursue the questions of the recruitment process in centromeric nucleosome formation at a structural level. To address these questions, they solved two cryo-EM structures: 1) centromeric nucleosome containing the native CEN3 DNA and 2) CBF3 core bound to the canonical nucleosome containing an engineered CEN3 DNA. They complemented these structures with modeling, and mutational studies.

The manuscript gives insights into the structure of CBF3 bound to nucleosomes. However the use of non-endogenous nucleosome substrates lowers the interest in this work. Additionally, the biochemical data is not sufficient to support their recruitment model and there is no cellular data. At this stage, without this data it is difficult to recommend this paper for publication.

Major points:

- The use of non-native substrates for complex formation is interesting but would require further validation. To obtain the structure of CEN3 CENP-ACse4 they used scFv for stabilization of nucleosomes and the structure of CBFcore was bound to nucleosome containing hybrid DNA and wt H3 histone. They provide reasons for this but why did they not try fixation which is very common these days and would allow them to determine the more relevant complexes?
- The binding curves in figure 3C are not saturated. Only NucCse4_L1LD/H2B/mut presents a beginning of saturation state. The authors should extend the titration range. Also, the Kd differences reported do not show a significant effect on the double mutant complex formation.
- The authors should comment how the position of the CEN3 core sequence affects the binding of CBFcore or Ndc10B?
- In figure 3B, CTF12 should show charge complementary or at least sequence registry.
- The clash score for the nucleosome - Gal4 complex and nucleosome - CBF3 complex structures are relatively high. This should be addressed, particularly at this resolution.

Minor

- The data is anisotropic, likely due to preferential orientation. Reprocessing/normalization is recommended.
- The authors should include a diagram of subunit composition of CBF3 for clarity.
- Since the phosphorylation state of CBF3 is necessary for CBF binding to nucleosomes, authors should attempt to find the phosphorylation site on the full-length protein.
- The NucCEN-601 substrate is heterogeneous as judged by the native gel (fig S2). Additional purification step may be required to improve the quality of binding data.

First of all, we would like to thank the reviewers for their detailed comments, which are very useful for improving the quality of our manuscript. We have revised the manuscript to address all of the concerns raised by the reviewers. The details of the revision are discussed in a point-to-point manner below.

Reviewer #1

The budding yeast centromere has been a powerful model for understanding eukaryotic centromeres for 40+ years, and is in the midst of important new structural insight from a handful of labs. Budding yeast is exceptional because DNA sequence specifies location of the centromere, unlike most eukaryotes where centromeres can exist on diverse DNA sequences. In this study, the authors solved structures of yeast CENP-ACSE4 nucleosomes on CEN3 DNA (bound by stabilizing antibody fragments) and CENP-ACSE4 nucleosomes on hybrid 601-CEN3 DNA bound by CBF3 (essential sequence specific binding complex). They also use biochemical approaches to address the role of CBF3 binding to CCG region on CEN3/601 DNA on CENP-A nucleosomes and oppose a model put forward of 'proximity looping' by the Barford lab in 2018. In the present study, the authors report selectivity of CBF3 core (without NDC10) over the full CBF3 complex and propose another mechanism for CBF3 interaction on nucleosomes. The authors also report structural details of CBF3 interactions on the centromeric nucleosome. Further, the authors propose the role/ mechanism of yeast genetic component CBF3 might act analogous to human genetic factor CENP-B in the CENP-A assembly and model of yeast CCAN assembly on CBF3 bound CENP-A nucleosomes. Overall the study has the potential to be of high interest throughout the centromere community, providing structural data that broadly addresses the role of inter-subunit interactions for CBF3 and CENP-A nucleosomes. I list several major and minor concerns that should be remedied before a revised version is considered for publication.

We appreciate reviewer #1 for the comment that "Overall the study has the potential to be of high interest throughout the centromere community, providing structural data that broadly addresses the role of inter-subunit interactions for CBF3 and CENP-A nucleosomes."

Major concerns:

1. The presentation of the position of the CENP-A nucleosome on CEN DNA is specifically confusing (it is also generally an insufficiently scholarly discussion; see point #2, below). The authors cite the positioning data coming from footprinting data in vivo (Xiao et al GD 2017 (ref 27)). But the sequence they use is quite shifted (by 12 nt if I am reading it correctly). The authors should explain any reason for these differences in the results section. Are there implications from their structure in relation to the 2017 paper regarding the sequence elements that Mif2 recognizes for its marked increased affinity for nucleosomes wrapped with CEN DNA? Does the dyad position impact on their model of CBF3-directed CENP-A nucleosome assembly?

We have clarified the issue of nucleosome positioning and its implications in the revised manuscript. We noted that the dyad position of the CEN3 Cse4 nucleosome used in

Xiao et al.'s paper (GD, 2017) is the result of computational modeling based on the hydroxyl radical foot printing data (Shaytan et al. *Nucleic Acids Res* 45: 9229-9243, 2017). As pointed out by reviewer # 1 correctly, the dyad position in their structural model is ~12 bp away from the dyad position in our structure. The difference shows that available knowledge on the determinant of nucleosome positioning is still limited. It is still not possible to predict nucleosome dyad position accurately for a given DNA sequence.

One implication of the difference in the dyad positions is that Mif2 would use its AT-hook region to interact both sides of the dyad to stabilize the nucleosome when the dyad position (nt 74) in our structure is used. Instead, using the dyad location at nt 61.5, Mif2 would appear to interact only one side of the dyad. Also, if dyad position at nt 61.5 is used, the CCG motif will not be sufficiently accessible for binding by the Gal4 domain due to the 12 bp shift of the dyad (or equivalent of shift of $12 - 10.5 = 1.5$ bp) considering the pitch of B DNA is 10.5 bp, i. e. CBF3 would not bind the nucleosome tightly using the dyad position at nt 61.5. These issues have been discussed in the revised manuscript (Top paragraph in page 10).

2. Nothing is cited for the central issue of mapping CENP-A nucleosome sequences in any species or for using those native sequences for understanding the role of DNA sequence for the structure and function of centromeric chromatin (except for one paper from this group). The native DNA sequence used here is a primary advance here relative to other yeast structural studies that primarily resort to the artificial 601 sequence, so the more important contributions from others in the field at large to this part of centromere studies really do need to be acknowledged in order for the reader to put the present findings in proper context with the published work.

In the revised manuscript, we expanded the introduction section and cited the earlier works that help define the centromeric DNA sequences. We also cited the papers that first identified Cse4 as the centromeric histone in budding yeast and structural studies of centromeric nucleosomes with the native and non-native sequences (see page 3 in the revised manuscript).

- >Manuelidis, L. Repeating restriction fragments of human DNA. *Nucleic Acids Res* **3**, 3063-76 (1976).
- >Manuelidis, L. & Wu, J.C. Homology between human and simian repeated DNA. *Nature* **276**, 92-4 (1978).
- >Waye, J.S. & Willard, H.F. Nucleotide sequence heterogeneity of alpha satellite repetitive DNA: a survey of alphoid sequences from different human chromosomes. *Nucleic Acids Res* **15**, 7549-69 (1987).
- >Clarke, L. & Carbon, J. Isolation of a yeast centromere and construction of functional small circular chromosomes. *Nature* **287**, 504-9 (1980).
- >Bloom, K.S. & Carbon, J. Yeast centromere DNA is in a unique and highly ordered structure in chromosomes and small circular minichromosomes. *Cell* **29**, 305-17 (1982).

- >Meluh, P.B., Yang, P., Glowczewski, L., Koshland, D. & Smith, M.M. Cse4p is a component of the core centromere of *Saccharomyces cerevisiae*. *Cell* **94**, 607-13 (1998).
- >Tachiwana, H. et al. Crystal structure of the human centromeric nucleosome containing CENP-A. *Nature* **476**, 232-5 (2011).
- >Dalal, Y., Wang, H., Lindsay, S. & Henikoff, S. Tetrameric structure of centromeric nucleosomes in interphase *Drosophila* cells. *PLoS Biol* **5**, e218 (2007).
- >Bui, M. et al. Cell-cycle-dependent structural transitions in the human CENP-A nucleosome in vivo. *Cell* **150**, 317-26 (2012).
- >Xiao, H. et al. Nonhistone Scm3 binds to AT-rich DNA to organize atypical centromeric nucleosome of budding yeast. *Mol Cell* **43**, 369-80 (2011).

3. What is implication for the need in this study to stabilize CENP-A nucleosomes by either an antibody fragment or 601 sequences introduced? Complementary experiments to understand what these artificial features 'do' to the sampling of different conformations/ states would be best. At a bare minimum, there should be some discussion what the need for these artificial stabilizing measures implies (as well as a discussion of the root cause of the instability by the extremely AT-rich CEN DNA sequence).

The main issue in cryo-EM study of nucleosome structures is that the surface tension at the water-air interface during the blotting and freezing process of the sample can disrupt the structure of the specimen. In the case of scFv₂-nucleosome complex, the antibody fragment (scFv) prevents the nucleosome from being exposed to the air-water interface, i.e. making the nucleosome stay in the water droplet. Instead of nucleosome, the scFv occupies the interface and got partially damaged (see Fig. S1). The "601" sequence, which is selected to bind the core histones more tightly, has higher affinity. The tighter

binding also helps resist the disruption of the nucleosome by the surface tension. According to current theory of nucleosome stability, the lack of GC in the major grooves of DNA that face away from the core histones in the nucleosome make the DNA unfavorable for bending, i.e. CEN3 DNA with AT-rich sequence is stiff, which makes it difficult to wrap the core histones. (see revision in page 5). The picture on the left shows the DNA sequence features that are favorable for binding of the core histones.

4. The authors mention on page 5 that core histones of CEN3 and 601 wrapped CENP-Acse4 nucleosomes have similar structures, but it is not clear the degree to which this has been measured. When aligning some landmark features of the nucleosome, what are the structural differences in these two structures? RMSDs?

We have provided more details for comparison of the two structures. Core histones in the two structures align very well with a backbone RMSD of 0.89 Å (page 6).

5. If CEN3 DNA is used in assay shown in Fig. 2B, is there similar binding with and without NDC10?

We have repeated the experiments in Fig. 2B two more times and quantified the bands. In comparison with CBF3core binding in Fig. 3B, we found that the CBF3 (with NDC10) binds to the Cse4 nucleosome with slightly weaker affinity in comparison with CBF3core (without NDC10) ($K_{d_{\text{CBF3core}}} = 0.32 \pm 0.06 \mu\text{M}$; $K_{d_{\text{CBF3}}} = 0.55 \pm 0.06 \mu\text{M}$). Note that the affinity difference is mainly caused by the missing of

~10% amplitude in the CBF3 binding curve, likely due to the non-specific binding of NDC10 to the nucleosome. The binding curve has been added to supplementary fig. 2h in the revised manuscript.

6. How robust is Fig. 2d regarding the 1 bp affects implied by the gel shown? Quantitation is not shown (as well as for some other of the gel-based experiments), so it is challenging to assess how much faith one has in this potentially striking result.

We have repeated the experiments in Fig. 2d two more times and quantified the results, confirming that 1 bp shift can have large effects on binding affinity. In addition, we have unpublished results about binding of a pioneer transcription factor to the nucleosome. 1 bp shift also show large effects on binding affinity. This is not surprising from structural point of view since 1 bp shift can lead to partial blocking of the binding site by the core histones in the nucleosome. The quantified result is added to the Fig. 2d.

7. How does the Gal4 position changes in the two conformations shown in Figure 2E? Can the authors report RMSD values? Does Gal4 bind to CCG region on CENP-A nucleosome DNA in both conformations?

Gal4 does not change its conformation relative to the nucleosome. The Gal4 domain and nucleosome form one rigid complex structure. The multiple conformations are the main core regions of CBF3 relative to the Gal4 domain-nucleosome. The Gal4 domain is linked to the major domain of Cep3 through a flexible linker (Supplementary Fig. 4b). We clarified this issue in the legend of Fig. 2e in the revised manuscript.

8. For Fig 3C, quantitation is shown, but there are very few details about this in the supplement (more details on this assay and the quantitation are needed). I infer that EtBr is used to assess bound/unbound, but DNA-binding proteins (and indeed nucleosomes) can potently restrict EtBr intercalation. Isn't there a better way to do this?

In the binding experiments, the gel shift experiments were done without EtBr. EtBr was only added after the gel shift experiment to stain the gel for detection. We quantified the free nucleosome (lane 1) and the nucleosome and CBF3core complex (lane 2 and 3) at the same concentration of nucleosome, the result shows that the ratio of integrated

intensity of the bands is 1.00: 1.01: 1.01, indicating that the EtBr have little effect on the

	Area	Mean	IntDen
1	92460	240.946	22277848
2	92460	238.946	22092937
3	92460	238.904	22089070

quantitation. This is reasonable because only the Gal4 domain binds to the CCG motif. In comparison of the total DNA length, the binding site is very small. The effects of histones on EtBr binding to DNA should not have affect our calculation since we are comparing free nucleosome and CBE3core-nucleosome complex. A more detailed description has been made in the supplementary method section in the revised manuscript.

9. It would clearer if the authors explicitly pointed out the differences between looping and proximity model along with their model shown in Figure 4a or 4c. Can the authors explain looping and proximity model in a similar manner as the cartoons are drawn in 4c? In the left panel, I would suggest that the authors show proximity models with dyad positions (along with their model) and indicate significant differences that would affect CBF3 mediated assembly of Cse4 during DNA replications in their structural model. Also, I was confused by the apparent symmetry of binding of the CCAN in their diagram. I thought that Xiao et al 2017 reported that the two copies of the CCAN component, CENP-C/Mif2, are bound asymmetrically with respect to the CEN DNA wrapped around the CENP-A nucleosome.

We thank review #1 for the suggestions. In the revised manuscript, we have drawn cartoons for the "looping" and "proximity" models (Fig 4c in revised manuscript). In the "proximity" model, Ndc10 is required to be close to Cse4-H4 in the nucleosome. In our model, Ndc10 does not coexist with the nucleosome structure as Ndc10 would sterically clash with nucleosomal DNA. The difference is mainly caused by the different positioning of the nucleosomes. In the "proximity" model, the dyad is speculated to be at nt 34, whereas it is at nt 74 in our structure.

Regarding the Xiao et al.'s result, Mif2 has three domains: the CENP-C motif, the AT-hook, and the dimerization domain. In Xiao et al.'s paper, they showed that the AT-hook domain of Mif2 bind to the AT-rich DNA on one side of the dyad (asymmetrically). CENP-C motif that binds to the acidic patch of H2A and the C-terminal tail of CENP-A on both sides of the nucleosome disk (symmetrically) in the structure of the nucleosome-CCAN by Yan et al. (Nature, 2019), the AT-hook domain is not observable. Figures on the left are

adopted from Xiao et al.'s paper. We have discussed these issues in page 10 in the revised manuscript.

10. Please add a reference for a study that demonstrates that the CCAN and CBF3 are bound to CSE4 nucleosomes simultaneously.

We have added three references.

A study by Ortiz et al. (Ortiz J, A putative protein complex consisting of Ctf19, Mcm21, and Okp1 represents a missing link in the budding yeast kinetochore. *Genes Dev.* 13:1140-1155, 1999.) suggests coexistence of Mif2, Cse4 and CBF3 complex, which involves the Ctf19-Mcm21-Okp1 complex.

Left is the diagram in their Figure 7 in Ortiz et al.'s paper.

It was also suggested by Yan et al.'s structural model (Yan et al. Structure of the inner kinetochore CCAN complex assembled onto a centromeric nucleosome. *Nature* 574, 278-282, 2019) in their extended Fig. 9d (left).

In addition, Ho et al. suggest that Kinetochore protein interactions with CBF3 are involved in the deposition of Cse4 (Ho et al. Localization and function of budding yeast CENP-A depends upon kinetochore protein interactions and is independent of canonical centromere sequence. *Cell Rep.* 9: 2027-2033, 2014).

Left is the graphical abstract of Ho et al.'s paper.

11. On page 10, the following sentence is very confusing: “The CEN3 CENP-ACse4 nucleosome-CBF3core complex is capable of co-binding of one copy of CCAN through the specific recognition of CENP-ACse4 by its subunit Mif227,32 in a way similar to the recognition of human CENP-A nucleosome by the CENP-A protein36.” The human CENP-A nucleosome can bind two copies of CENP-C. Is the point here about the stoichiometry of components? If so, it isn’t correct then. If not, then the meaning of the sentence and the reasoning for citing the human arrangement is unclear.

We apologize for the confusion. We have divided this long sentence into two short sentences for clarity in the revised manuscript. The CEN3 CENP-A^{Cse4} nucleosome-CBF3core complex is capable of co-binding of one copy of CCAN. The specific recognition of CENP-A^{Cse4} by CCAN was achieved by the CENP-C motif in its subunit Mif2/CENP-C (see question 9 for discussion of Mif2 binding to the CENP-A nucleosome).

12. The idea to have a closing discussion on the relation of their findings involving CBF3 to what happens in species (like humans) with CENP-B could be potentially useful. But exclusively discussing the similarities, as the authors do here, is misleading. Yes,

CENP-B provides some centromere function, but unlike CBF3 (or other key molecules like CENP-A/Cse4 and CENP-C/Mif2) it is dispensable for cell and organismal viability. Also, it is true that the initial types of human artificial chromosomes required CENP-B, but the latest generation of human artificial chromosomes completely bypass the need for CENP-B by seeding centromeric nucleosomes. To my knowledge, nobody is considering a scenario where CBF3 is bypassed for budding yeast centromere identity and/or function. That all said, while CBF3/CENP-B is not a perfect comparison, a somewhat extended discussion on this to close the paper would be very helpful if both the key similarities and differences are included.

We have extended our discussion to clarify the concern raised by reviewer #1. In addition to pointing out the similar role of CBF3 and CENP-B in recruiting Cse4 and H3.3 by DNA-sequence dependent manner, we noted that the functional importance of CBF3 and CENP-B is different. CENP-B is dispensable for cell and organismal viability. It is likely epigenetics plays a major role in human centromere function.

Minor concerns:

13. Cite Supplementary figure 2a in Paragraph 1 in page 6.

Supplementary figure 2a is cited in paragraph 1 on page 7 in the revised manuscript.

14. Supplementary figures 2e, f are not referenced in the text.

They have been rearranged as Supplementary figs 2g, h and cited in page 7 in the revised manuscript.

Reviewer #2:

The authors pursue the questions of the recruitment process in centromeric nucleosome formation at a structural level. To address these questions, they solved two cryo-EM structures: 1) centromeric nucleosome containing the native CEN3 DNA and 2) CBF3 core bound to the canonical nucleosome containing an engineered CEN3 DNA. They complemented these structures with modeling, and mutational studies.

The manuscript gives insights into the structure of CBF3 bound to nucleosomes.

We appreciate reviewer #2 for the comment that “The manuscript gives insights into the structure of CBF3 bound to nucleosomes.”

However, the use of non-endogenous nucleosome substrates lowers the interest in this work. Additionally, the biochemical data is not sufficient to support their recruitment model and there is no cellular data. At this stage, without this data it is difficult to recommend this paper for publication.

It would have been ideal to have solved the structure of the Cse4 nucleosome bound to CBF3core and to provide cellular data to further support our model. However, we believe our manuscript with the current structural and biochemical data is still suitable for a publication in *Nature Communications* for the following reasons.

(1) We solved the structure of the CEN3 Cse4 nucleosome. This is a long-sought structure in the budding yeast centromere field. We have published a paper on the structural determination of a native-like human CENP-A nucleosome structure alone without additional biochemical and cellular data in *Nature Communications* (Zhou et al. Atomic resolution cryo-EM structure of a native-like CENP-A nucleosome aided by an antibody fragment, 10:2301, 2019).

(2) Our binding studies used the nucleosome containing Cse4 protein (Fig. 3c), which confirms the structural model for the Cse4 nucleosome bound to CBF3core (Fig. 3b): CBF3core has weak, dynamic interactions with core histones.

(3) Our manuscript is focused on addressing the structural issues of budding yeast centromeric nucleosome formation. Our diagram shown in Fig. 4d in the revised manuscript is analogous to the "looping" and "proximity" models (Fig. 4c in the revised manuscript). They are structure-based hypotheses/speculations that need to be tested in future studies. Both the "looping" and "proximity" models were published in *Nature Structure and Molecular Biology* without additional cellular data to support them.

(4) A test of the model with cellular data would be beyond the scope of this manuscript and our expertise. It would require the amount of work that likely deserves a publication itself. Instead, we have found some in vivo studies in the literature that are consistent with the events described in Figure 1d. For example, our model suggests that CBF3core and CCAN could co-bind to the Cse4 nucleosome. Ortiz et al. has shown that Cse4, CBF3 complex and Mif2 can coexist in vivo and Ho et al. showed that CBF3 and kinetochore can coexist for delivering Cse4 to form the nucleosome (see our answers to question 10 by reviewer #1). Also, consistent with the dissociation of Ndc10 after the formation of the centromeric nucleosome, Ndc10 is found to be enriched at the spindle midzone in late anaphase (Bouck, D.C. & Bloom, K.S. The kinetochore protein Ndc10p is required for spindle stability and cytokinesis in yeast. *Proc Natl Acad Sci U S A* **102**, 5408-13,2005).

Major points:

- The use of non-native substrates for complex formation is interesting but would require further validation. To obtain the structure of CEN3 CENP-ACse4 they used scFv for stabilization of nucleosomes and the structure of CBFcore was bound to nucleosome containing hybrid DNA and wt H3 histone. They provide reasons for this but why did they not try fixation which is very common these days and would allow them to determine the more relevant complexes?

Our mutational and binding studies used the nucleosome containing Cse4 histone (Fig. 3c), which confirms the structural model that shows CBF3core make only weak dynamics interactions with core histones (Fig. 3b, Fig. 2e, and supplementary Fig. 4e).

We tried cross-linking fixation first but failed. And we believe other groups (David Barford at MRC and Stephen Harrison at Harvard) also tried unsuccessfully. Otherwise, they would have used native CEN DNA instead of Widom 601 DNA. So far, only our antibody approach is successful in determination of the structures of nucleosomes containing native-DNA sequences by cryo-EM. In the revised manuscript, we have stated the failure of cross-linking approach (see page 5).

- The binding curves in figure 3C are not saturated. Only NucCse4_L1LD/H2B/mut presents a beginning of saturation state. The authors should extend the titration range. Also, the Kd differences reported do not show a significant effect on the double mutant complex formation.

We have added the data points that extend the titration range to show saturation in the revised Fig. 3c.

The small effects on binding affinity from mutation studies are consistent with the cryo-EM structures that CBF3core only interact the core histones dynamically, i. e. only a small fraction of the population of the CBF3core-nucleosome complex involves close interactions between CBF3core and core histones.

- The authors should comment how the position of the CEN3 core sequence affects the binding of CBFcore or Ndc10B?

We thank reviewer #2 for the suggestion. We have discussed this issue in the revised manuscript (page 11). CEN3 DNA can position the nucleosome uniquely. The position of the CEN3 DNA in our cryo-EM structure allows CBF3core to bind the nucleosome while Ndc10B would clash with the nucleosomal DNA. In contrast, in the "proximity" model, the nucleosome with the dyad at nucleotide 34 is not an intrinsically favored nucleosome position and Ndc10B would not clash with the nucleosome.

- In figure 3B, CTF12 should show charge complementary or at least sequence registry.

Based on our model of CEN3 Cse4 nucleosome, we observed putative salt bridge and hydrophobic interactions. These interactions are shown in Supplementary Fig. 4c, along with local density map at the interface between Ctf13 and core histones.

- The clash score for the nucleosome - Gal4 complex and nucleosome - CBF3 complex structures are relatively high. This should be addressed, particularly at this resolution.

We have made further refinement of the structures, which reduced the clashscores of the two complexes substantially. The new results are shown in the revised Table 1 and the validation reports.

Minor

- The data is anisotropic, likely due to preferential orientation. Reprocessing/normalization is recommended.

We have tried to reprocess/normalize the cryo-EM data by reducing the number of particles with side views while trying to add more those with top views for the CBF3core-nucleosome complex. However, due to the limited total number of particles, it did not improve the quality of the density map. In principle, we could collect more particles and then do the normalization but the COVID situation limits our access to the Krios. Since improvement of the quality of the density map will not change our conclusion, we have kept the initial density map in the revised manuscript.

- The authors should include a diagram of subunit composition of CBF3 for clarity.

A diagram has been provided for the subunits of CBF3core in Fig. 2f to show the composition of CBF3core.

- Since the phosphorylation state of CBF3 is necessary for CBF binding to nucleosomes, authors should attempt to find the phosphorylation site on the full-length protein.

The phosphorylation site is in the L1 loop of Skp1, which is described by Leber et al. (Leber, V., Nans, A. & Singleton, M.R. Structural basis for assembly of the CBF3 kinetochore complex. *EMBO J* **37**, 269-281, 2018). The Skp1 L1 loop sequence HDSNLQNNSDSESDSDSETNHKSKDNNN (residue 37-64) includes several ser residues. We showed that the Skp1 mutant with the deletion of this loop is sufficient for binding of CBF3core to the nucleosome (Supplementary Fig. 2h).

- The NucCEN-601 substrate is heterogeneous as judged by the native gel (fig S2). Additional purification step may be required to improve the quality of binding data.

We tried to purify the NucCEN-601 nucleosome further but found that the minor bands above the major band were caused by the native page gel for the nucleosome containing CEN3-601 DNA.

Reviewers' Comments:

Reviewer #1:

Remarks to the Author:

The authors have improved the manuscript in some places, with attention given to many of the reviewer comments. These improvements include additional experiments and analysis that improve the clarity and rigor of the study. There are three areas, however, where the authors' responses are disappointing.

1. Their response to my Major Comment #2 falls short because it does not cite papers from human nor yeast on mapping CENP-A nucleosome positions on centromere DNA using next generation DNA sequencing methodologies. Rather, there are citations added for the identifying centromere DNA itself, a study by Tachiwana et al (2011) that uses the incorrect human position for generating their palindromic crystallographic DNA template, and -- most mystifyingly -- references for older controversial proposals for half nucleosomes that have been debunked by several groups. The debunking includes evidence from the Wu and Bai labs, themselves, so I found this particularly confusing. Relating their claims to yeast centromere nucleosome mapping studies using conventional nucleosome positioning, as well as the Widom trick with a chemically reactive histone H4, and perhaps other approaches, must be done. Those studies are directly relevant to their new proposal for the dyad positioning site, as they have expanded upon in the revised manuscript while addressing my Major Comment #1. All of this needs substantial improvement in the text to clearly relate their findings to the relevant literature.

2. For my concerns about their DNA sequence claims, as well as for a related point from Reviewer #2 (his/her first Major point), the authors seem unaware that a high-resolution cryo-EM structure of the human centromeric nucleosome on natural centromere DNA was published by another group (Allu et al., 2019, Current Biology) contemporaneous with their 2019 antibody approach. The Current Biology study didn't require an antibody fragment and the centromere nucleosome was reported to tolerate routine crosslinking used in cryo-EM studies. So there is a lack of clear, valid reasoning in the authors' responses on these points.

3. While they did improve the last part of the discussion by adding the functional distinction between CBF3 and CENP-B, they did not provide references for the findings that support this important distinction.

Reviewer #2:

Remarks to the Author:

The authors addressed my questions in the revised manuscript. I suggest publication.

Response to reviewers' comments

Reviewer #1:

The authors have improved the manuscript in some places, with attention given to many of the reviewer comments. These improvements include additional experiments and analysis that improve the clarity and rigor of the study. There are three areas, however, where the authors' responses are disappointing.

We thank reviewer #1 for critical reading of our revised manuscript.

1. Their response to my Major Comment #2 falls short because it does not cite papers from human nor yeast on mapping CENP-A nucleosome positions on centromere DNA using next generation DNA sequencing methodologies. Rather, there are citations added for the identifying centromere DNA itself, a study by Tachiwana et al (2011) that uses the incorrect human position for generating their palindromic crystallographic DNA template, and -- most mystifyingly -- references for older controversial proposals for half nucleosomes that have been debunked by several groups. The debunking includes evidence from the Wu and Bai labs, themselves, so I found this particularly confusing.

The Major Comments #2 by reviewer #1 states that "Nothing is cited for the central issue of mapping CENP-A nucleosome sequences in any species or for using those native sequences for understanding the role of DNA sequence for the structure and function of centromeric chromatin (except for one paper from this group). The native DNA sequence used here is a primary advance here relative to other yeast structural studies that primarily resort to the artificial 601 sequence, so the more important contributions from others in the field at large to this part of centromere studies really do need to be acknowledged in order for the reader to put the present findings in proper context with the published work."

Our earlier understanding of the Major Comments #2 is that it is about the earlier studies of the centromeric DNA sequences and the structures of the centromeric nucleosomes. Therefore, we cited the papers that describe the initial discovery of the human and centromeric DNA sequences, and structural studies of centromeric nucleosomes. The goal was to provide the background information without our own judgement on issues that were debated. We apologize for the misunderstanding.

Relating their claims to yeast centromere nucleosome mapping studies using conventional nucleosome positioning, as well as the Widom trick with a chemically reactive histone H4, and perhaps other approaches, must be done. Those studies are directly relevant to their new proposal for the dyad positioning site, as they have expanded upon in the revised manuscript while addressing my Major Comment #1. All of this needs substantial improvement in the text to clearly relate their findings to the relevant literature.

In our previous revision, we have cited earlier studies on the mapping of centromeric nucleosome positions in budding yeast, which includes the following:

Xiao, H. et al. Molecular basis of CENP-C association with the CENP-A nucleosome at yeast centromeres. *Genes Dev* **31**, 1958-1972 (2017).

Krassovsky, K., Henikoff, J.G. & Henikoff, S. Tripartite organization of centromeric chromatin in budding yeast. *Proc Natl Acad Sci U S A* **109**, 243-8 (2012).

Cole, H.A., Howard, B.H. & Clark, D.J. The centromeric nucleosome of budding yeast is perfectly positioned and covers the entire centromere. *Proc Natl Acad Sci U S A* **108**, 12687-92 (2011).

Furuyama, S. & Biggins, S. Centromere identity is specified by a single centromeric nucleosome in budding yeast. *Proc Natl Acad Sci U S A* **104**, 14706-11 (2007).

In this second revision, we also included the paper from Henikoff lab that studied budding yeast centromeric nucleosome in vivo using the Widom trick with a chemically reactive histone H4 in the literature, and the study on human centromeric nucleosomes from Black lab.

Henikoff, S. et al. The budding yeast Centromere DNA Element II wraps a stable Cse4 hemisome in either orientation in vivo. *Elife* **3**, e01861 (2014).

Hasson, D. et al. The octamer is the major form of CENP-A nucleosomes at human centromeres. *Nat Struct Mol Biol* **20**, 687-95 (2013).

2. For my concerns about their DNA sequence claims, as well as for a related point from Reviewer #2 (his/her first Major point), the authors seem unaware that a high-resolution cryo-EM structure of the human centromeric nucleosome on natural centromere DNA was published by another group (Allu et al., 2019, Current Biology) contemporaneous with their 2019 antibody approach. The Current Biology study didn't require an antibody fragment and the centromere nucleosome was reported to tolerate routine crosslinking used in cryo-EM studies. So there is a lack of clear, valid reasoning in the authors' responses on these points.

We thank reviewer #1 for pointing out this study. We apologize for this omission and have added this reference and others published very recently in the revised manuscript. The structure by Allu et al. from Ben Black group is about the complex of human CENP-A nucleosome bound to CENP-C and CENP-N proteins fixed by chemical cross-linking. As we pointed out to reviewer #2, cross-linking approach is not applicable for the budding yeast centromeric nucleosome. Also, the resolution of Allu et al.'s structure is relatively low on the DNA part. Therefore, it is still true that the scFv-assisted cryo-EM approach is so far the most effective approach for solving atomic resolution structures of the nucleosomes with native-like DNA sequences. Nevertheless, this point is not important for the current manuscript. Our reply to reviewer #2 is simply to point out that cross-linking method did not work for the budding yeast centromeric nucleosomes, likely due to its very weak binding caused by the AT-rich sequences.

3. While they did improve the last part of the discussion by adding the functional distinction between CBF3 and CENP-B, they did not provide references for the findings that support this important distinction.

We apologize for this omission. In the revised manuscript, we have cited the recent paper from Ben Black's group that shows CENP-B is not essential for generating human artificial chromosomes.

Logsdon, G.A. et al. Human Artificial Chromosomes that Bypass Centromeric DNA. *Cell* **178**, 624-639 e19 (2019).